# Non-Contact, Simple Neonatal Monitoring by Photoplethysmography

**DOI:** 10.3390/s18124362

**Published:** 2018-12-10

**Authors:** Juan-Carlos Cobos-Torres, Mohamed Abderrahim, José Martínez-Orgado

**Affiliations:** 1Postgraduate Subdirection, Catholic University of Cuenca, Cuenca 4X8V+5F, Ecuador; 2Department of Systems Engineering and Automation, University Carlos III of Madrid, Leganes 28911, Spain; mohamed@ing.uc3m.es; 3Neonatology, San Carlos Clinical Hospital, Madrid 28040, Spain; jose.martinezo@salud.madrid.org

**Keywords:** image processing, neonatology, imaging photoplethysmography, vital signs, respiratory sinus arrhythmia

## Abstract

This paper presents non-contact vital sign monitoring in neonates, based on image processing, where a standard color camera captures the plethysmographic signal and the heart and breathing rates are processed and estimated online. It is important that the measurements are taken in a non-invasive manner, which is imperceptible to the patient. Currently, many methods have been proposed for non-contact measurement. However, to the best of the authors’ knowledge, it has not been possible to identify methods with low computational costs and a high tolerance to artifacts. With the aim of improving contactless measurement results, the proposed method based on the computer vision technique is enhanced to overcome the mentioned drawbacks. The camera is attached to an incubator in the Neonatal Intensive Care Unit and a single area in the neonate’s diaphragm is monitored. Several factors are considered in the stages of image acquisition, as well as in the plethysmographic signal formation, pre-filtering and filtering. The pre-filter step uses numerical analysis techniques to reduce the signal offset. The proposed method decouples the breath rate from the frequency of sinus arrhythmia. This separation makes it possible to analyze independently any cardiac and respiratory dysrhythmias. Nine newborns were monitored with our proposed method. A Bland-Altman analysis of the data shows a close correlation of the heart rates measured with the two approaches (correlation coefficient of 0.94 for heart rate (HR) and 0.86 for breath rate (BR)) with an uncertainty of 4.2 bpm for HR and 4.9 for BR (k = 1). The comparison of our method and another non-contact method considered as a standard independent component analysis (ICA) showed lower central processing unit (CPU) usage for our method (75% less CPU usage).

## 1. Introduction

There are many different invasive and non-invasive methods of measurements in the context of neonatal pathology and therapeutic interventions. Among the non-invasive techniques for heart rate monitoring, the worldwide gold standard, is the electrocardiograph (ECG). However, recording the electrical potential generated by the heart requires proper electrode placement, which may interfere with the baby’s movements and interaction with parents and/or caretakers.

Surprisingly, even today disadvantaged areas still lack quality electrodes, complicating their placement, causing skin lesions. Additionally, electrode misplacement may produce skin lesions or ischemia if wrapped around a limb. These disadvantages can lead to a work overload for health personnel who must revise the electrodes regularly [1].

Another important measurement method is the finger pulse oximeter with a plethysmography wave; it measures heart rate (HR)and oxygen saturation in the blood but can cause problems like those caused by an ECG electrode including skin irritations. New oximeters also provide breath rate (BR) information linked to respiratory sinus arrhythmia. The third method for measuring heartbeat and breathing frequencies is based on the piezoelectric effect and involves placing a sensor on the abdominal area [2]. The above methods are the most commonly used for monitoring vital signs and require contact sensors to be placed on the patient, whereas new trends seek to allow non-contact monitorization.

Newer methods for measuring heart and respiration rates include thermal imaging analysis [3,4], observation of the Doppler effect [5], Doppler-camera hybrid [6], and imaging photoplethysmography. The last technique analyzes changes in the color intensity and optical properties of a selected area of the skin that is affected by variations in blood flow or by pulsating vibrations of vital signs.

Robust methods have been developed to cope with variations in light intensity but they can still be affected by the type of light source or flickering. These methods are called photoplethysmographic images (PPGI). One method that works under ambient lighting focuses on the human face but can only provide adequate results with stable illumination [7]. A more robust method was based on source separation through the application of an independent component analysis (ICA) [8]. It has a high computational cost and the original signals must be independent; something which does not happen if breath rate is measured based on sinus arrhythmia.

Another relevant method is Eulerian magnification [9], which uses an extra layer of pre-processing that amplifies the redness of the person’s skin in order to measure the HR. This also carries a heavy computational cost and requires an initial measuring point to provide a more precise approximation. An HR extraction system was built and tested using video footage of three hospitalized neonates [9,10]. The system produced a good measurement of HR from real-world videos taken under good lighting conditions and when the subjects’ movements were smooth; nevertheless, the degree of agreement or concordance (e.g., the Bland-Altman method) between PPGI measurements and the pulse oximetry measurements was not provided.

After reviewing the principal PPGI methods, we considered their suitability for monitoring in a Neonatal Intensive Care Unit (NICU) environment. This is important because possible areas or regions for measurements have not been determined. Additionally, newborns’ movements are gentle and subtle but they may also have spasms that generate artifacts or false measurements. Therefore, such movements can affect the accuracy of photoplethysmographic measurements. In addition, NICU lighting may be different in every hospital. Consequently, below we describe different PPGI methods that have been implemented in NICUs.

The Mestha method [11] for continuous HR monitoring has a low accuracy in environments with excessive movement. This is due to the use of a high pass finite impulse response (FIR) filter to remove very low-frequency components. The FIR filters do not work properly with low frequencies and are not optimal given the large number of coefficients they require. Additionally, the published study performed measurements in an environment with excellent stable lighting (300 lux), so the results cannot demonstrate that the method would work properly under poor conditions.

Another method based on an ICA analysis has been presented by Scalise [12] but is also characterized by a high computational cost. Moreover, in that report, the light conditions were stable and always under control according to the authors. The authors affirm that the camera was placed perpendicularly in front of the neonate’s face at a distance of 20 cm. However, this is not reflected in the example of the picture provided and some light reflections are visible on the transparent wall of the incubator.

The method based on joint-time-frequency diagrams (JTFD) presented by Aarts [13] reports an accurate measurement of the plethysmographic signal but no details were given regarding signal processing or filtering. In addition, the method is not robust when faced with artifacts produced by motion and changes in illumination.

The Wang method [14] introduces a mathematical model. They analyze the relevant optical and physiological properties of skin reflections. This is important in order to increase the understanding of the algorithmic principles behind PPGI. However, the PPGI mathematical model is only based on the assumption of a single light source with a constant spectrum.

Finally, the method based discrete wavelet decomposition with a periodicity-based voting scheme [15]. This method relies on the convolution of the original signal with FIR filter structures. As already explained, FIR filters do not work properly with low frequencies.

After reviewing a number of the methods developed for PPGI, we have designed a robust and efficient method that works under changing light conditions and in the presence of moderate motion artifacts. In addition, the system has a computationally lightweight algorithm that is appropriate for online use. Finally, the method can be implemented as part of a monitoring system in a NICU environment.

The method presented in the current work is based on the analysis of color intensity variations in a given area on the newborn infant’s diaphragm. A pre-filtering step based on numerical analysis techniques was adopted to reduce the offset of the averaged signal. The proposed non-invasive approach employs a standard camera and image processing to estimate the desired variables. In addition, the method for BR measurement is decoupled from the sinus arrhythmia frequency.

## 2. Materials and Methods

The overall scheme adopted for the processing of the images and estimation of physiological variables is illustrated in Figure 1. The method consists of taking the images from a video camera, pre-processing them by selecting an area of interest and separating each image into its color components. The selection of the area of interest is done manually. Next, the pixels in each frame are averaged. Three stacks are generated with each averaged value and with a given length (last six seconds of the video). These signals are filtered while the stacks continue to be updated. The filter has two steps: an offset filter and narrow-band filters. The filtered signal is analyzed through a short-time Fourier transform (STFT) (to find the frequency components of the signal).

The first segment in Figure 1 corresponds to the video capture in the area of analysis and the details of this process are given in Section 2.2. The second segment presents signal formation for the analysis. The third segment represents the steps to filter the signal image and then extract the frequencies of interest. A description of the processes involved in segments two and three is given in Section 2.3 and Section 2.4. The fourth segment summarizes the steps for signal processing and estimation of the physiological variables. The steps in the method are explained further in the next sections of the paper.

### 2.1. Patient Population

The study was performed in accordance with the Declaration of Helsinki regarding studies on human subjects and was approved by the ethics committee of the Hospital Vicente Corral Moscoso. The newborns who took part in this study, with the informed consent of their parents, were nine preterm infants with a gestation of 25–40 weeks, a weight of 500 g or greater, and without any complications. Patient safety was never compromised because the medical protocols and/or protective measures were not changed. Several recordings of each newborn were made and stored for later analysis in segments. To explain the proposed method, data from only one of the newborns have been used. All of the newborns’ samples were used to validate the method with the Bland-Altman technique. During the measurements, newborns were inside the incubator and the camera was placed at an approximate distance of 50 cm.

### 2.2. Set-Up and Experimental Procedures

The experimental setup consisted of a digital camera placed outside the glass of the incubator directly targeting the newborn and a second camera focusing on the vital signs monitor, to check and synchronize the variables being measured, as shown in Figure 2.

Measurements were made in the NICU of the University Hospital Puerta de Hierro-Majadahonda and the Hospital Vicente Corral Moscoso. The only light sources were the natural light from windows and skylights in the roof and/or artificial light from fluorescent lamps. The video sequences of the neonates were recorded with a digital camera. The resolution of the videos is 1920 × 1080 or 1280 × 720 pixels and the frame rate is 24 or 30 frames per second (fps). Image sequences were saved in AVI or MP4 format. The measurements corresponding to the vital signs were collected from the vital signs monitor. As a reference, the second camera recorded the vital signs from the monitor screen, heart rate and respiration rate. The two cameras were paired to start the recordings simultaneously. A simple script in Matlab analyzed the videos frame by frame. The script stored the readings at a frequency of 30 Hz. The values are averaged every second. The averaged value will be the reference value to be compared with our proposed method. The NICU monitor was Dräguer Infinity Delta.

### 2.3. Pre-Processing

The images from the different video segments were exported to MATLAB^®^ for post-processing and pre-filtering. The first step was to select the area of interest for analysis. The BR was visible in the abdominal and thoracic movements of newborns [16]. The abdominal area was chosen because it contains information reflecting the heartbeat and breath rates. These two vital signs cause changes in the intensity level of pixel brightness in the images. The measurements of these two signs are independent and unrelated to the sinus arrhythmia.

The area in question was 40 × 40 pixels, from which the image was separated into three RGB-channels. The photoplethysmographic information contained in this area was sufficient to measure the vital signs. The input R-channel had higher intensities; it did not provide more plethysmographic information but it was important for analysis in poor light conditions. The RGB intensity value ranged from 0–255 for each of the RGB components of each pixel in the color image. Since not all of the pixels in the selected region of interest contained a variation in the brightness of the plethysmographic signal, the pixels were combined into a single average brightness value, Equations (1)–(3).
(1)$$averageR=\frac{1}{wh}\sum_{x=1}^{w}\left[\sum_{y=1}^{h}R_{xy}\right]$$
(2)$$averageB=\frac{1}{wh}\sum_{x=1}^{w}\left[\sum_{y=1}^{h}B_{xy}\right]$$
(3)$$averageG=\frac{1}{wh}\sum_{x=1}^{w}\left[\sum_{y=1}^{h}G_{xy}\right].$$

The steps of this process are summarized schematically in Figure 3.

Additionally, as shown in Figure 4a, the image for the region of interest (ROI) analysis; Figure 4b the neonate in the incubator with a mark for the analysis area (red box); Figure 4c, the offset in the signal was very high, due to the illumination, pigmentation of the skin, and the distance from the camera. This offset was filtered and treated as part of the low-frequency components. The signal trend was calculated using numerical methods. First, the straight line that best adjusts to the signal was obtained by least squares. Then, the trend signal was generated through a moving average.

### 2.4. Post-Processing

The first second of the video segment was removed because the first moments are always unstable in a video recording (the images are brighter than normal). After calculating the average for the ROI and each color channel, FIFO stacks (First in, first out) were generated. These stacks had an equivalent of 6 s of information. This period was selected to obtain a minimum of three heartbeats and three breaths. While a stack continued to be updated, the low-frequency signals were filtered from the full stack. The sources of these signals depend on the distance from the camera as well as the lighting of the skin. These low frequencies were estimated using numerical analysis techniques to reduce the displacement signal. First, the least squares method was applied to find the linear function that best fits the signal. This function was subtracted from the original signal to reduce the offset. Second, large variations were smoothed by calculating quotients with the signal deviations. After obtaining the pre-filtered signal, two narrow band filters (type infinite impulse response (IIR)) attenuated the frequencies outside the band of interest. These bands are detailed in Table 1.

The method finds continuous the HR and BR measurements by applying STFT and peak detection at small time intervals that coincide with the maximum 160 bpm for HR. This is equivalent to a period of 0.375 s, so a FIFO stack was created with the last samples from the signal. A delay was made so that there were at least three peaks for the respiratory rate in the signal. Consequently, a six-second delay at the start was found to produce a good accuracy for the estimation of the frequency and a continuous reading in the detection of peaks in the signals.

The STFT was used to obtain the spectrum in one of the 6 s sliding-windows that the average signal runs (the sliding-window moved in 0.1 s steps). The spectrum of one of these sliding-windows is shown in Figure 5. The identified frequency peaks can be noted in the filtered signal. The narrow-band filters are the IIR. The Butterworth filter was chosen because the bandwidths required a lower order. This meant that fewer calculations were needed and decreased the computational cost.

Finally, the position of the highest peak was identified within the band of vital signs and the values were printed. If the value of any of the vital signs were close to any pre-set limit, an alarm was activated.

## 3. Results

The validation and information on the performance and reliability of the system were analyzed with the Bland-Altman method [18]. The differences between estimates from contact and remote measurements were plotted against the averages of both systems for HR and BR, Figure 6a,b and Table 2, respectively. In the HR case, the measurements were recorded every second. In contrast, the BR was recorded every three seconds, thus, an over correlation was avoided by the repetition of values. Means are represented by dotted lines; 95% limits of agreement (±1.96 media (SD)) are represented by dashed lines on the plots in Figure 6. Specifically, the mean biases were −1.5 bpm with 95% limits of agreement −9.7 to 5.8 bpm for the HR and −0.6 bpm with 95% limits of agreement −9.2 to 10.3 bpm for the BR. The standard deviation of the residuals was 4.5 bpm for HR and 4.97 bpm for BR. In addition, the estimated and reference data were correlated. Respectively, 360 and 120 pairs of measurements from nine videos for HR and BR are plotted in Figure 6c,d.

A Pearson’s coefficient was also calculated for the heart and breath rates, with respective ranges of 0.94 for HR and 0.86 for BR, confirming the correlation between the two HR and BR measurements. There was a slight systematic difference between the two measurements. The various monitoring apparatus used in the different rooms had different error ranges (some had an accuracy of ±3 percent according to their manufacturer). In addition, atypical values caused by strong movements of the baby’s abdominal area, shadows cast by medical personnel, or changes in the light intensity were visible. The lighting in some rooms, which was a combination of artificial light (fluorescent lamps) and natural light (windows and skylights in the ceiling) resulted in different shadows.

The degree of agreement for the measurements was calculated using the measurements taken for 120 s, as shown in Figure 6c,d. The figures show that the results with our method were comparable with those obtained by the NICU monitor. Results are shown in Table 3.

CPU usage was recorded when both scripts worked at the same time. Scripts analyzed and processed the same information. In order to show the lightweight of our algorithm, we have registered the CPU usage history through data comparison graph. The computer was an Intel(R) Core(TM) i7-4500U CPU (1.8 GHz, 4 MB cache).

The results of this exercise are shown in Figure 7 and Table 4. In terms of CPU usage, our method used an average of about 3.00% while the other method used an average of 12.2%, which means that our method requires almost 75% less of CPU usage.

In addition, to test the robustness of our algorithm, we implemented the algorithm developed by Poh et al. [8], which is considered as a standard in image-based photoplethysmography to perform a comparison. This algorithm was chosen based on their high performance as reported by the authors. To make an objective evaluation of each approach in terms of its HR estimation accuracy, both methods were executed to analyze the same ROI from the patient video. These measurements allow the two methods to be evaluated while they were operating in a changing environment (variations in light type, intensity when the subject is interacting with another person for more 3 min). The measurement is evaluated in Figure 8.

All the statistical analysis, tables and graphical methods have performed with the MedCalc^®^ statistical software.

## 4. Discussion

The analysis carried out by Verkruysse [7], citing Reference [19], indicates that the G-channel provides the strongest plethysmographic signal, which corresponds to an oxyhemoglobin absorption peak, but the study also indicates that RB-channels contain plethysmographic data. The abundance of capillaries, tissue perfusion, hemoglobin oxygenation and skin thickness can cause rosacea, reddishness or paleness, or a cyanotic tone, which are more visible in white skinned people. Good tissue perfusion with adequate levels of oxygenated red blood cells produces a pink color in the skin.

When analyzing the images, we found that the skin showed a pink hue in low light and was pale with more light; however, the red component always showed a high-intensity level. This is important since light-dark alternations are a powerful and important signal to achieve the temporal organization of the circadian rhythm in newborns as well as promote their weight gain and growth while hospitalized [20]. Therefore, work on R-channel analysis is essential, since NICU illumination is not constant.

The information received from the camera is a reflection of hemoglobin oxygenation. Figure 9 shows that the R-channel intensity maintains a high-level independently of any variations in illumination. In addition, in Figure 4c higher intensity is observed in the R-channel when compared to GB-channels for a good level of patient oxygenation (about twice the average intensities of the R-channel, with respect to the other two channels). When working with a dim light source, the idea is to start with the assumption of a less accurate measurement on the G-channel given the higher absorption by this component. In this case, the absence of light reflection indicates a low level of patient oxygenation.

Nevertheless, all of this depends on the type of camera used. Specifically, the sensor and the filter installed for each color channel. The majority of cameras obtain more information from the G-channel [21] because the human eye is more sensitive to this color and the cameras emulate this sensitivity. In any case, due to the low computational cost, the three channels can be processed with the method presented here.

The results presented here show that non-contact supervision through computer vision techniques may be included in future modalities for neonatal monitoring in hospital NICUs. Very high-risk patients requiring close specialized supervision, such as those with heart problems, would still need an ECG to collect the necessary information.

Infants who need special care and are in an incubator have not achieved control of their nervous system. Their movements are spasmodic with small tremors in the hands or legs. As the nervous system matures and muscle control improves, jolts and tremors change to gentler movements of the legs and arms. These movements will not affect the measurements of vital signs by plethysmography. More mature infants have greater psychomotor control and may make movements that the system will not be able to filter properly. This could be corrected by implementing a movement tracking system and combining it with the method proposed here. The method to reduce the offset signal is very efficient, and the whole system can be run online.

Studies of sudden infant death syndrome (SIDS), whose cause is still unknown have observed that apnea may precede bradycardia [22]. The British programs to control for SIDS use transcutaneous oxygen monitors because they can decouple the cardiorespiratory variables on the monitoring equipment [23,24]. This makes it possible to detect signs of hypoxia or obstructive apnea. Therefore, it is important to emphasize that our procedure has the advantage that the measurement is made without referring to sinus arrhythmia and disassociates the cardiorespiratory variables. Cardiac and respiratory dysrhythmias can be analyzed independently.

Figure 10 depicts a special case that we obtained during a measurement. The vital signs monitor developed in Matlab shows an abrupt HR decrease (3–4 s) that is interpreted by the doctor as a benign paroxysmal supraventricular tachycardia. It is a relatively frequent phenomenon in children with no pathological history of any kind. Signs of heart failure should be ruled out especially in younger patients and in those who have more hours of clinical evolution [25]. This heart rate arrhythmia was not repeated in the measurements.

At present, the results obtained by PPGI are not completely categorized; the measurements still lack reliable validation protocols. What is noteworthy is that the measurement by PPGI in neonates differs from classic detection methods, because it will not cause discomfort, injuries, etc.

The reason for choosing the abdominal area as the analysis region is that breathing in newborn infants is visible in abdominal and thoracic movements, which makes it easy to distinguish. In addition, measurement also depends on the viewing angle; different test runs showed that the best placement for the camera was to attach it to the transparent wall of the incubator, pointed at the baby’s diaphragm. This resulted in an optimum viewing angle and produced a better quality measurement of physiological variables.

Finally, it will be necessary to relax the cut-off frequencies to allow for a larger range of physiologic measurements. Our proposed method could be useful for sick newborns. The heart and breathing rates could go outside of these bounds.

## 5. Conclusions

This paper describes a methodology to register cardiac and respiratory rates from video recordings of the abdominal area of newborns in NICUs. To the best of our knowledge, this is the first demonstration of a relatively low-cost method for neonatal physiological measurement that permits an independent analysis of cardiac and respiratory rates; also, this method is imperceptible to the patient. The structure and algorithm of this system have not yet been tested online within a NICU setting. We have worked with videos because NICU access is limited so as to avoid disturbing the patients. In order to evaluate the robustness and lightweight of our algorithm, we have performed experiments to compare it with one of the most cited existing image photoplethysmography methods. The experiment proved that our method is more robust and computationally more efficient.

The improvement of this system can be achieved. This method would not work in the case of poor lighting or darkness. The system could be improved by dedicated illumination and optimal lighting conditions that would decrease or avoid shadows. In addition, working in parallel with distance measurement systems, such as the Kinect camera, is expected to result in better concordance among measurements and decrease in artifacts effects.

On the other hand, with this approach, the authors have shown how easy it is to apply the system in existing incubators. We believe it has a real potential to expand and improve access to health care. In addition, this methodology should respond adequately to the increasing demand for home hospitalization and telemedicine in the near future. Given the low cost and wide availability of portable devices with integrated cameras, this technology is also promising for basic residential monitoring of children by their parents. Of course, measurement must be made in alternative areas of analysis, such as the child’s face to extract the HR. Moreover, it could be a practical solution for measuring BR, but it needs to incorporate an adequate motion tracking feature. Although the scope of the work described in this document is limited to recovery of the heart and respiratory rates, many other important physiological parameters can be measured using the same video images. Therefore, this line of research should be continued with the aim of developing a real-time and multi-node system for measuring physiological variables.

In summary, our contributions are:By applying numerical analysis techniques, the authors have established a filter for reducing very low frequencies. This reduces the artifacts and the offset of the plethysmographic signal.Three color channels are used to analyze the plethysmographic signal. This allows a more robust estimation of HR and BR because its measurements can be interpolated.The developed algorithm is simple and of low computational cost, which makes it adequate for implementation in portable devices with low computing power.The sinus arrhythmia is not taken for the measurements by the method itself, which is important to supply an independent analysis of the cardiac and respiratory dysrhythmias rhythms.With all the above, a robust and complete system has been developed, which is adequate for monitoring neonates in a NICU environment.

## Figures and Tables

**Figure 1 sensors-18-04362-f001:**
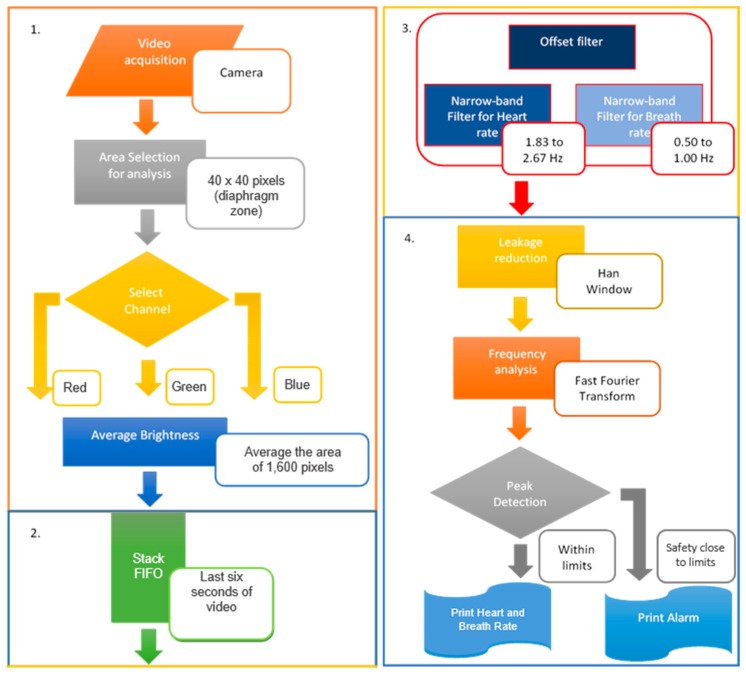
Sequence used for measurement of heartbeat and breathing.

**Figure 2 sensors-18-04362-f002:**
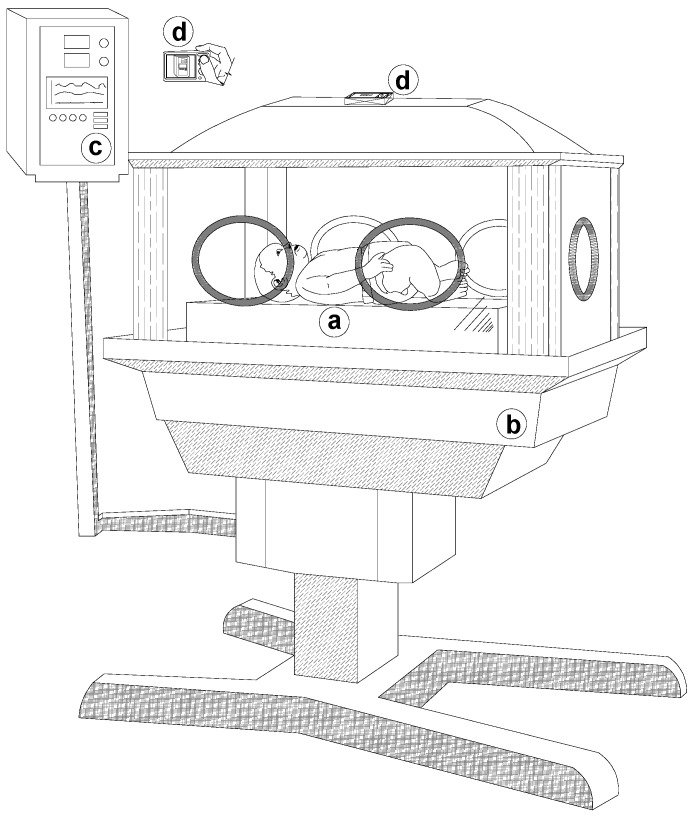
Setup of the experiment: (**a**) neonate, (**b**) Neonatal Intensive Care Unit (NICU Incubator, (**c**) vital signs monitor, and (**d**) cameras.

**Figure 3 sensors-18-04362-f003:**
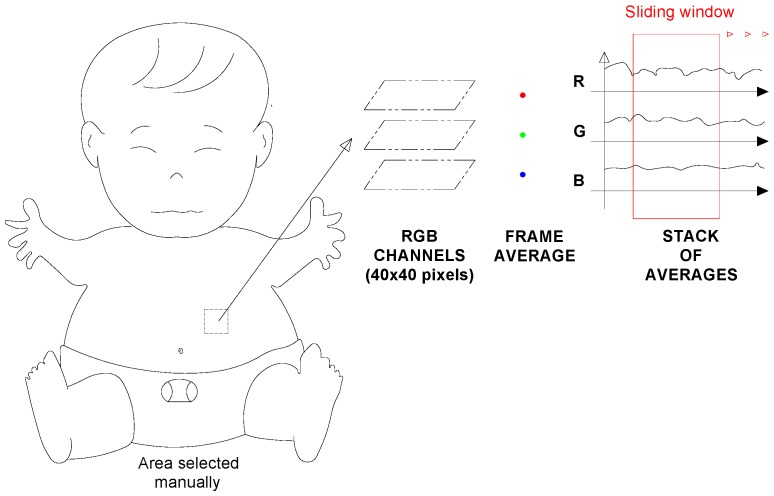
Illustration of processing: area selection, separation of RGB-channels, an average of each signal.

**Figure 4 sensors-18-04362-f004:**
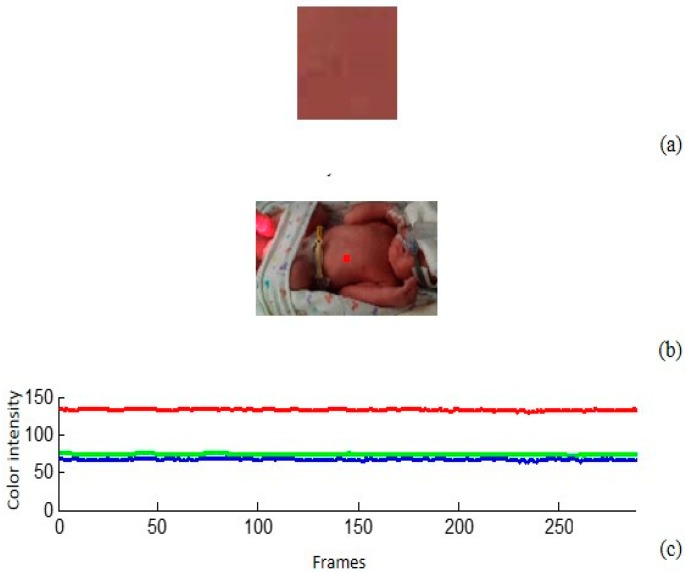
Capture of the plethysmographic signal processing with Matlab, (**a**) region of interest (ROI), (**b**) image analysis, and (**c**) graph average signals of the RGB-channels.

**Figure 5 sensors-18-04362-f005:**
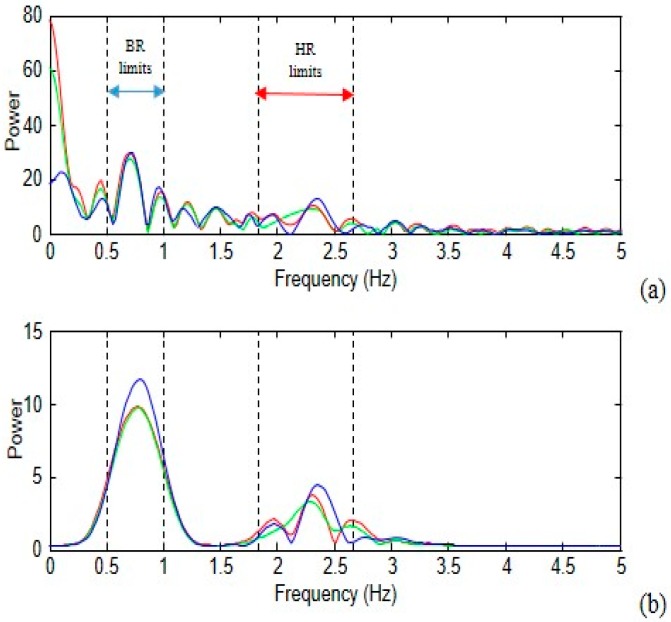
Spectrum analysis of the three RGB-channels, (**a**) unfiltered spectrum and (**b**) filtered spectrum, a sliding-windows of 6 s (the dashed lines are the limits for the heart and breath rates).

**Figure 6 sensors-18-04362-f006:**
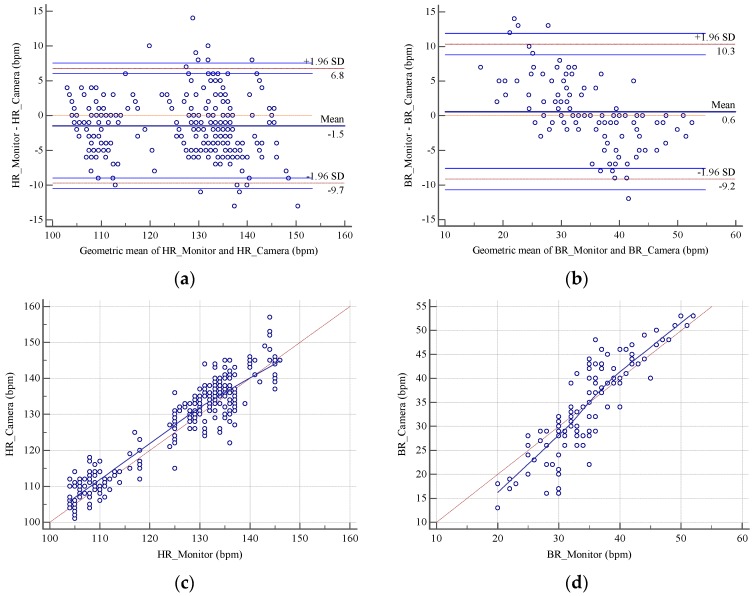
Bland-Altman plot showing the level of agreement between (**a**) HR monitor and HR with the camera, (**b**) BR monitor and BR with the camera. Dispersion diagram showing the relationship between the measurements made with the contact method and the proposed method (**c**) HR monitor and HR with the camera, (**d**) BR monitor and BR with the camera.

**Figure 7 sensors-18-04362-f007:**
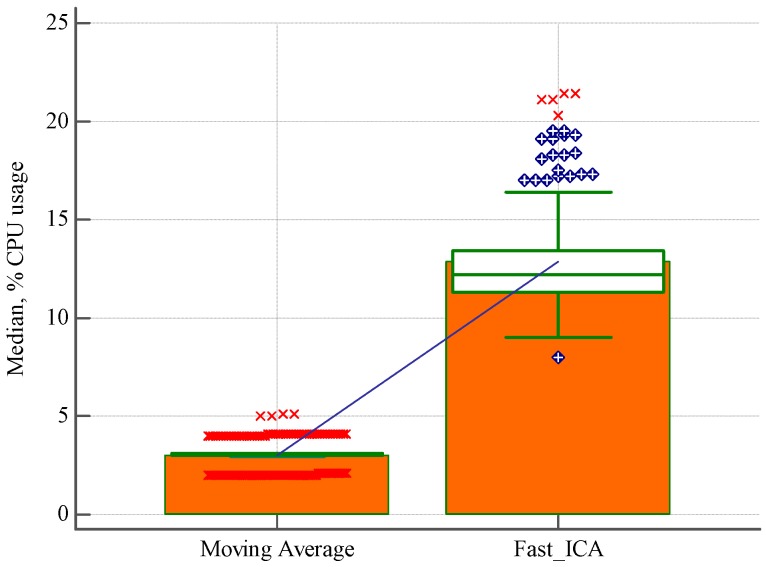
Performance comparison between our method and independent component analysis (ICA).

**Figure 8 sensors-18-04362-f008:**
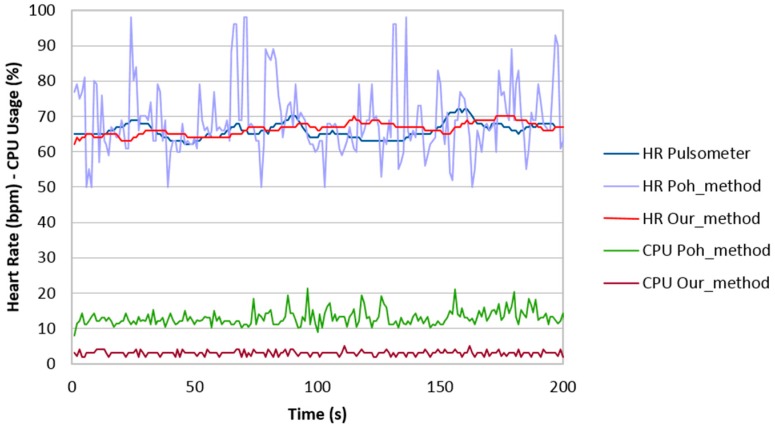
Performance comparison between our method and ICA.

**Figure 9 sensors-18-04362-f009:**
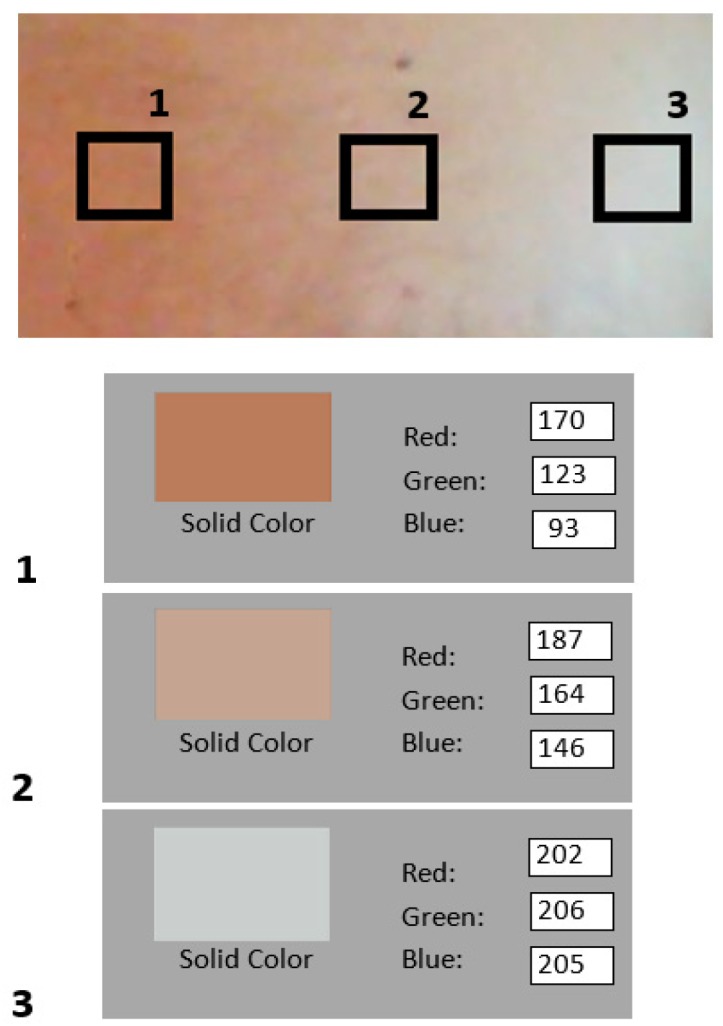
Comparison of intensities in RGB-channels, with different levels of illumination.

**Figure 10 sensors-18-04362-f010:**
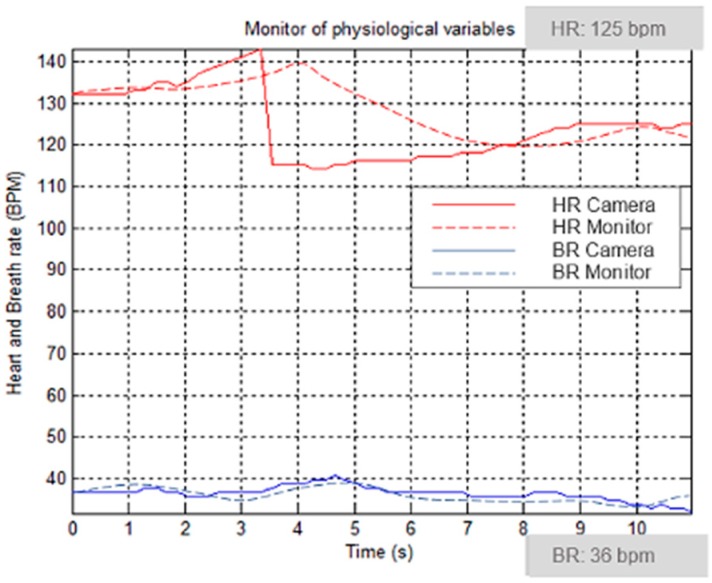
Signal Monitor development in Matlab vs. NICU monitor.

**Table 1 sensors-18-04362-t001:** Band of interest.

Vital Signs	Lower Limit (bpm)	Upper Limit (bpm)
HR	110 (1.83 Hz)	160 (2.67 Hz)
BR	30 (0.50 Hz)	60 (1 Hz)

Hear rate (HR) and breath rate (BR) limits for healthy newborn infants [17].

**Table 2 sensors-18-04362-t002:** Information on the differences found in the measurements by the Bland-Altman method.

	HR (bpm)	BR (bpm)
**Sample size**	360	120
**Arithmetic mean**	−1.4667	0.5917
**Standard deviation**	4.2135	4.9747
**Lower limit**	−9.7252	−9.1588
**Upper limit**	6.7918	10.3421

**Table 3 sensors-18-04362-t003:** Concordance correlation coefficients.

Parameters	HR (bpm)	BR (bpm)
**Sample size**	360	120
**Concordance correlation coefficient**	0.9342	0.7988
**95% Confidence interval**	0.9198 to 0.9461	0.7402 to 0.8454
**Pearson ρ (precision)**	0.9410	0.8585
**Bias correction factor Cb (accuracy)**	0.9927	0.9305

**Table 4 sensors-18-04362-t004:** CPU usage comparison graph.

Parameters	Moving Average	Fast_ICA
**Sample size**	360	120
**Lowest value (%)**	2.0	8.0
**Highest value (%)**	5.1	21.4
**Median (%)**	3.0	12.2
**25th percentile (%)**	3.0	11.3
**75th percentile (%)**	3.1	13.4

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
