# Peer review of "Non-Contact, Simple Neonatal Monitoring by Photoplethysmography"

_sensors, 2018, doi:10.3390/s18124362_

Round 1
Reviewer 1 Report
Your article describes a study using your non-contact photoplethysmographic imaging system to monitor the heart rate and breathing rates in a newborn population. As it currently stands, the manuscript is incomplete as it does not provide sufficient description of your study and additional analyses are needed to demonstrate the performance of your approach and support the claims you are reporting about your approach. I have the following major and minor comments that I hope help you to revise your work.
Major Comments:
1. Your study design as described in Sections 2.1 and 2.5 is incomplete. It does not provide an adequate description of how the heart rate and breathing rate measurements from your device were compared to the reference measurements. It also does not state what devices (including manufacturer and model) were used to record the reference measurements. In Figure 4 you show a camera taking pictures of the vital signs monitor. This seems to suggest that you used this images of the vital signs monitor to compare with your devices measurements. You should provide additional information on how these measurements were taken and how the timing of measurements from the various devices was aligned. This should describe how long and frequently measurements were taken, how measurements were paired between the devices, and how (if) data was averaged over time from the camera measurements to compare with the reference measurements.
2. Figure 7 shows the Bland-Altman plots between your device and reference measurements. This information alone is an incomplete representation of the performance of your device compared to the reference measurements. You should also provide scatter plots of the monitor and compare values for both heart rate and breathing rate along with linear regression analyses to demonstrate that the slope is close to 1 and the offset close to 0 between the different measurement sources. You should also provide the mean, standard deviation, minimum and maximum value for the heart rate and breathing rates from the reference and camera measurements to enable the reader to understand the range of data that the results represent. A statistical analysis section should also be added describing how your comparisons were made.
3. Your estimation of heart and breathing rates seems to come from frequency analysis of the band-pass filtered RGB signals. The performance of measurements close to the lower and upper cutoff frequencies will surely be affected by the filter designs. You list low and upper limits for healthy newborns for the two physiologic parameters in Table 1 and then seem to use these values as the cut-off frequencies. This is not an acceptable approach for multiple reasons. 1) Using the physiologic limits as the cut-off frequencies will distort measurement performance when the true measurements are close to these values. 2) These are the values for healthy newborns, but your technology may be most useful for sick newborns whose heart and breathing rates may go outside of these bounds. I would recommend you relax these cut-off frequencies to allow for a larger range of physiologic measurements. Otherwise, you should provide a rationale of why these cut-off frequencies are adequate for your device. It may be necessary to identify a different data processing approach that is able to account for more variations in the physiologic range if you expect this to be useful in clinical populations.
4. One of your claims in your abstract is that your approach has less computational costs than other approaches. In your conclusion you also state that you compared your approach with another imaging photoplethysmographic method. The only place either of these results seem to be shown are in Figure 9 of your Discussion section from a single patient. This single plot does not support your statement related to lower computational costs or performance comparison to another method. You should perform this analysis on all available data if you wish to make this claim and also performance statistical analyses to demonstrate that it is true.
5. The current organization of your manuscript makes it difficult to follow. Please reorganize your manuscript to keep all pieces of the technology description, study design, and results in one place. For example, your study design is partly described in Section 2.1 and partly described in Section 2.5. Other statements in the results and discussion add more pieces of the study design. Please consider the following when revising your manuscript:
a. Sections 2.1 and 2.5 should be adjacent to each other as these both describe your study design.
b. Section 2.4 does not seem to describe your method and should be moved to the discussion.
c. Lines 250-262 in Section 3. Results seem to describe your method and should be grouped with the sections describing your technology.
d. Lines 331-340 and Figure 9 in the Section 4. Discussion seem to describe results. These should be reported in the Results section.
Minor Comments:
1. Your abstract only makes vague reference to your results with statements such as ‘good estimates were obtained’. You should provide more specific information about the study design and results in the abstract.
2. In the first paragraph of the introduction you refer to ECG as the gold standard for respiration rate measurement. This is not true. The best reference standard for respiration rate would be respiratory gas monitoring through capnography.
3. Line 54: I believe ‘photoplethysmography’ should be ‘imaging photoplethysmography’ to refer to non-contact camera methods.
4. Much of the introduction is vague descriptions of other methods. This could be reduced to make the introduction more concise, readable, and focus on the points important to your current study.
5. Lines 135-141: I believe the subsections B, C, D that are referred to should be sections 2.2, 2.3, 2.4.
6. Figure 8 and the description in Lines 310-315 are not clear. It seems out of place and isn’t event clear to me if this is from your data or taken from something else. This should be removed or the significance highlighted with more explanation.
7. Line 321-322: This line lacks detail and just refers to things it seems you are also working on. It doesn’t seem to be necessary to include this in the manuscript.
8. Line 366: The line ‘it could be a practical solution for controlling BR’ is very unclear. I’m not sure what you are trying to say with this, but I don’t think that you mean you could use your approach to control a patient’s breathing rate which is what the statement currently suggests.
Author Response
RESPONSE TO REVIEWER 1
Comments and Suggestions for Authors:
Your article describes a study using your non-contact photoplethysmographic imaging system to monitor the heart rate and breathing rates in a newborn population. As it currently stands, the manuscript is incomplete as it does not provide sufficient description of your study and additional analyses are needed to demonstrate the performance of your approach and support the claims you are reporting about your approach. I have the following major and minor comments that I hope help you to revise your work.
Authors’ Reply:
We thank the reviewer for these positive, insightful, and constructive comments, which have helped us improve the quality of the paper dramatically.
Major Comments:
Comment 1.
Your study design as described in Sections 2.1 and 2.5 is incomplete. It does not provide an adequate description of how the heart rate and breathing rate measurements from your device were compared to the reference measurements. It also does not state what devices (including manufacturer and model) were used to record the reference measurements. In Figure 4 you show a camera taking pictures of the vital signs monitor. This seems to suggest that you used this images of the vital signs monitor to compare with your devices measurements. You should provide additional information on how these measurements were taken and how the timing of measurements from the various devices was aligned. This should describe how long and frequently measurements were taken, how measurements were paired between the devices, and how (if) data was averaged over time from the camera measurements to compare with the reference measurements.
Authors’ Reply: Thanks for pointing this out. We apologize for the incomplete description. We hope that the following paragraph clarifies how the measurements were compared.
“As a reference, the second camera recorded the vital signs from monitor screen, heart rate and respiration rate. The two cameras were paired to start the recordings simultaneously. A simple script in Matlab analyzed the videos frame by frame. The script stores the readings at a frequency of 30 Hz. The values are averaged every second. The averaged value will be the reference value to be compared with our proposed method. The NICU monitor are Dräguer Infinity Delta.”
Comment 2.
Figure 7 shows the Bland-Altman plots between your device and reference measurements. This information alone is an incomplete representation of the performance of your device compared to the reference measurements. You should also provide scatter plots of the monitor and compare values for both heart rate and breathing rate along with linear regression analyses to demonstrate that the slope is close to 1 and the offset close to 0 between the different measurement sources. You should also provide the mean, standard deviation, minimum and maximum value for the heart rate and breathing rates from the reference and camera measurements to enable the reader to understand the range of data that the results represent. A statistical analysis section should also be added describing how your comparisons were made.
Authors’ Reply: We appreciate his observations. We have provide scatter plots of the monitor and compare values for both heart rate and breathing rate along with linear regression analysis (Figure 6 (c) and Figure 6 (d)). Besides, we have provide information on differences found in the measurements by the Bland-Altman method (Table 2) and concordance correlation coefficients (Table 3).
Comment 3.
Your estimation of heart and breathing rates seems to come from frequency analysis of the band-pass filtered RGB signals. The performance of measurements close to the lower and upper cutoff frequencies will surely be affected by the filter designs. You list low and upper limits for healthy newborns for the two physiologic parameters in Table 1 and then seem to use these values as the cut-off frequencies. This is not an acceptable approach for multiple reasons. 1) Using the physiologic limits as the cut-off frequencies will distort measurement performance when the true measurements are close to these values. 2) These are the values for healthy newborns, but your technology may be most useful for sick newborns whose heart and breathing rates may go outside of these bounds. I would recommend you relax these cut-off frequencies to allow for a larger range of physiologic measurements. Otherwise, you should provide a rationale of why these cut-off frequencies are adequate for your device. It may be necessary to identify a different data processing approach that is able to account for more variations in the physiologic range if you expect this to be useful in clinical populations.
Authors’ Reply: Your assessment is correct, but the experiments were carried out in this way. The distortion near the limits is not high. When the frequencies approach the limits, the alarms are working. In any case, in the discussion section, we have added the following paragraph.
“Finally, it will be necessary relax the cut-off frequencies to allow for a larger range of physiologic measurements. Our proposed method could be useful for sick newborns. The heart and breathing rates could go outside of these bounds.”
Comment 4.
One of your claims in your abstract is that your approach has less computational costs than other approaches. In your conclusion, you also state that you compared your approach with another imaging photoplethysmographic method. The only place either of these results seem to be shown are in Figure 9 of your Discussion section from a single patient. This single plot does not support your statement related to lower computational costs or performance comparison to another method. You should perform this analysis on all available data if you wish to make this claim and also performance statistical analyses to demonstrate that it is true.
Authors’ Reply: Excellent your observation, we have performed this analysis with all available data (Figure 7). Besides, the statistical analysis are performed (Table 4).
Comment 5.
The current organization of your manuscript makes it difficult to follow. Please reorganize your manuscript to keep all pieces of the technology description, study design, and results in one place. For example, your study design is partly described in Section 2.1 and partly described in Section 2.5. Other statements in the results and discussion add more pieces of the study design. Please consider the following when revising your manuscript:
a. Sections 2.1 and 2.5 should be adjacent to each other as these both describe your study design.
b. Section 2.4 does not seem to describe your method and should be moved to the discussion.
c. Lines 250-262 in Section 3. Results seem to describe your method and should be grouped with the sections describing your technology.
d. Lines 331-340 and Figure 9 in the Section 4. Discussion seem to describe results. These should be reported in the Results section.
Authors’ Reply: We have organized the manuscript to keep all pieces of the technology description, study design, and results in one place. We have considered all observations. We appreciate your professional review.
a. Sections 2.1 and 2.5 are contiguous
b. Section 2.4 was moved to Discussion section.
c. Lines 250-262 and Figure 6 were grouped in subsection 2.4
d. Lines 331-340 and Figure 9 were reported in the Results section.
Minor Comment 1.
Your abstract only makes vague reference to your results with statements such as ‘good estimates were obtained’. You should provide more specific information about the study design and results in the abstract.
Authors’ Reply: We have provide specific information about results in the abstract.
“Bland-Altman analysis of data show close correlation of the heart rates measured with the two approaches (correlation coefficient of 0.94 for HR and 0.86 for BR) with an uncertainty of 4.2 bpm for HR and 4.9 for BR (k=1). The comparison of our method and another non-contact method considered as a standard (ICA) showed lower CPU usage for our method (75% less CPU usage).”
Minor Comment 2.
In the first paragraph of the introduction you refer to ECG as the gold standard for respiration rate measurement. This is not true. The best reference standard for respiration rate would be respiratory gas monitoring through capnography.
Authors’ Reply: Your observation is correct. We have eliminated the respiratory rate in this statement.
“There are many different invasive and non-invasive methods of measurement in the context of neonatal pathology and therapeutic interventions. Among non-invasive techniques for heart rate the worldwide gold standard is the electrocardiograph (ECG).”
Minor Comment 3.
Line 54: I believe ‘photoplethysmography’ should be ‘imaging photoplethysmography’ to refer to non-contact camera methods.
Authors’ Reply: It is correct, we have change the term.
“Newer methods for measuring heart and respiration rates include thermal imaging analysis [3], [4], observation of Doppler Effect [5], Doppler-camera hybrid [6], and imaging photoplethysmography.”
Minor Comment 4.
Much of the introduction is vague descriptions of other methods. This could be reduced to make the introduction more concise, readable, and focus on the points important to your current study.
Authors’ Reply: In a respectful way we do not think so. We have detailed:
- the Importance of the subject,
- the conceptual and historical background of the subject and
- the definition of the problem.
All with current and novel bibliographical support.
Minor Comment 5.
Lines 135-141: I believe the subsections B, C, D that are referred to should be sections 2.2, 2.3, 2.4.
Authors’ Reply: This is correct. We have changed it.
“The first segment in Figure 1 corresponds to the video capture in the area of analysis, and the details of this process are given in subsection 2.2. The second segment presents signal formation for analysis. The third segment represents the steps to filter the signal image and then extract the frequencies of interest. The description of the processes involved in segments two and three is given in subsections 2.3 and 2.4. The fourth segment summarizes the steps for signal processing and estimation of the physiological variables. The steps in the method are explained further in the next sections of the paper.”
Minor Comment 6.
Figure 8 and the description in Lines 310-315 are not clear. It seems out of place and isn’t event clear to me if this is from your data or taken from something else. This should be removed or the significance highlighted with more explanation.
Authors’ Reply: Again, your observation is very important. We have improved the explanation.
“Figure 10 depicts a special case that we obtained during a measurement. The vital signs monitor developed in Matlab shows an abrupt HR decrease (3 - 4 seconds) that is interpreted by the doctor as a benign paroxysmal supraventricular tachycardia. It is a relatively frequent phenomenon in children with no pathological history of any kind. Signs of heart failure should be ruled out especially in younger patients and in those who have more hours of clinical evolution [25]. This heart rate arrhythmia was not repeated in the measurements.”
Minor Comment 7.
Line 321-322: This line lacks detail and just refers to things it seems you are also working on. It doesn’t seem to be necessary to include this in the manuscript.
Authors’ Reply: You are absolutely right, we have eliminated it.
Minor Comment 8.
Line 366: The line ‘it could be a practical solution for controlling BR’ is very unclear. I’m not sure what you are trying to say with this, but I don’t think that you mean you could use your approach to control a patient’s breathing rate which is what the statement currently suggests.
Author's Reply: We have written the paragraph wrongly. We hope that the idea is now clear.
“Moreover, it could be a practical solution for measuring BR, but it needs to incorporate an adequate motion tracking feature”
Reviewer 2 Report
Dear Authors,
Below are my comments regarding your paper. These comments need to be addressed before further processing:
1. There are too many paragraphs describing the background and again brought up throughout the paper. This makes it difficult to read and appreciate the outcomes the authors are trying to establish. Relevant and focussed background is always necessary, but it must be streamlined here.
2. The selection of ROI is manual; was the ROI manually selected for each of the frames in the videos? If so this needs to be addressed with the ROI being selected automatically.
3. Why is the area of interest chosen as 40x40 pixels? Is there any specific reason?
4. The threshold of the lower and upper limit should have been fixed by a medically authorised device or by doctors.
5. What would happen in the case of night-time ,with no lighting or in the case of IR; how would results in the night or variations in daylight change the scenario. The author mentioned in the page 7 line 229 that the “only light source were the natural light from window…” What will be the result variation in the night and day scenario?
6. The mathematical model behind the signal processing must be included to provide readers an understanding by the authors of the working principles of the system.
Best wishes,
Author Response
RESPONSE TO REVIEWER 2
Comments and Suggestions for Authors:
Below are my comments regarding your paper. These comments need to be addressed before further processing:
Authors’ Reply: We appreciate your time and dedication. Your comments are very valuable to improve our paper.
Comment 1.
There are too many paragraphs describing the background and again brought up throughout the paper. This makes it difficult to read and appreciate the outcomes the authors are trying to establish. Relevant and focussed background is always necessary, but it must be streamlined here.
Authors’ Reply: We apologize. The first reviewer has recommended that we rearrange the text. We hope that this new structure facilitates the paper reading.
Comment 2.
The selection of ROI is manual; was the ROI manually selected for each of the frames in the videos? If so this needs to be addressed with the ROI being selected automatically.
Authors’ Reply: It must be a bad interpretation. We specify the manual selection of the ROI (line 129 and Figure 3).
Comment 3.
Why is the area of interest chosen as 40x40 pixels? Is there any specific reason?
Authors’ Reply: Your doubt is right. We have tried to clarify this with the following sentence:
“The area in question is 40 x 40 pixels, from which the image is separated into three RGB-channels. The photoplethysmographic information contained in this area is sufficient to measure the vital signs.”
Comment 4.
The threshold of the lower and upper limit should have been fixed by a medically authorized device or by doctors.
Authors’ Reply: Thanks for your suggestion but the limits were set by the collaborating doctor.
Comment 5.
What would happen in the case of night-time ,with no lighting or in the case of IR; how would results in the night or variations in daylight change the scenario. The author mentioned in the page 7 line 229 that the “only light source were the natural light from window…” What will be the result variation in the night and day scenario?
Authors’ Reply: We have tried clarifying this in the following sentence:
"The improvement of this system can be achieved. This method would not work in the case of poor lighting or darkness.The system could be improved by dedicated illumination and optimal lighting conditions that would decrease or avoid shadows."
Comment 6.
The mathematical model behind the signal processing must be included to provide readers an understanding by the authors of the working principles of the system.
Authors’ Reply: You appreciation is very perceptive but with all due respect, we think that least squares and moving average are standard numerical analysis techniques, it would not be very appropriate to put their mathematical models.
Reviewer 3 Report
Please refer and compare to other recent literature concerning video based HR monitoring based on ICA, i.e. using Hilbert-Huang-Transform.
Author Response
RESPONSE TO REVIEWER 3
Comments and Suggestions for Authors:
Please refer and compare to other recent literature concerning video based HR monitoring based on ICA, i.e. using Hilbert-Huang-Transform.
Authors’ Reply: Thanks for your recommendation. After reviewing the paper, we are convinced that you think that both methods are similar. Yes, both are non-parametric methods to represent a trend. Better, we appreciate your recommendation. We have begun to compare both nonparametric methods. This will be a next publication.
Reviewer 4 Report
The manuscript by Cobos-Torres et al proposed a non-contact way to measure photoplethysmograph with colour camera for neonatal monitoring. With imaging processing and some signal processing technique, the heart rate and respiratory rate were further extracted for the evaluation of the health status of neonate. The study would be of highly interest to the audience of the journal. However, the method for non-contact vital signs measurement is not new. The authors need to highlight the major contribution that has not been addressed before. This manuscript can be further improved with the following comments:
1. In the introduction, the motivation and the research gap between current study and previous research is not clear. Please further clarify it.
2. As far as I am concerned, the normal respiratory frequency for the neonate is between 0.5 to 1.0 Hz. To detect the abnormal range, spectrum that is out of the [0.5, 1.0] Hz should also be considered.
3. Section 2.1, “… the distance of 50 cm”. Please state the reference point for the distance measurement.
4. Please state how long was the video measured. With only 6 seconds signals being extracted, it might be not enough to measure the respiratory rate.
5. In the Fig. 7, what is the y-axis represents for? What is the unit for both x and y-axis?
6. Since performance of the proposed method, either the computational cost, or the resistance to the motion artefact is the major motivation for this study, please include the analysis and results in the results part, instead of in the discussion part.
Author Response
RESPONSE TO REVIEWER 4
Comments and Suggestions for Authors:
The manuscript by Cobos-Torres et al proposed a non-contact way to measure photoplethysmograph with colour camera for neonatal monitoring. With imaging processing and some signal processing technique, the heart rate and respiratory rate were further extracted for the evaluation of the health status of neonate. The study would be of highly interest to the audience of the journal. However, the method for non-contact vital signs measurement is not new. The authors need to highlight the major contribution that has not been addressed before. This manuscript can be further improved with the following comments:
Authors’ Reply: We appreciate the above supportive comments.
Comment 1
In the introduction, the motivation and the research gap between current study and previous research is not clear. Please further clarify it.
Authors’ Reply: In a respectful way we do not think so. We have detailed:
- the Importance of the subject,
- the conceptual and historical background of the subject and
- the definition of the problem.
All with current and novel bibliographical support.
Comment 2
As far as I am concerned, the normal respiratory frequency for the neonate is between 0.5 to 1.0 Hz. To detect the abnormal range, spectrum that is out of the [0.5, 1.0] Hz should also be considered.
Authors’ Reply: Thanks for your suggestion but the limits were set by the collaborating doctor. In any case, your doubt is complemented by a comment from the first reviewer.
For this reason, in the discussion section, we have added the following paragraph.
“Finally, it will be necessary relax the cut-off frequencies to allow for a larger range of physiologic measurements. Our proposed method could be useful for sick newborns. The heart and breathing rates could go outside of these bounds.”
Comment 3
Section 2.1, “… the distance of 50 cm”. Please state the reference point for the distance measurement.
Authors’ Reply:
Comment 4
Please state how long was the video measured. With only 6 seconds signals being extracted, it might be not enough to measure the respiratory rate.
Authors’ Reply: It seems a bad interpretation. The measurements were made in times of 40 seconds. The 6 seconds correspond to the sliding window.
Comment 5
In the Fig. 7, what is the y-axis represents for? What is the unit for both x and y-axis?
Authors’ Reply: Thanks, for you observation. It has already been done.
Comment 6
Since performance of the proposed method, either the computational cost, or the resistance to the motion artefact is the major motivation for this study, please include the analysis and results in the results part, instead of in the discussion part. .
Authors’ Reply: Again, we appreciate your accurate comments. We have already done this restructuring of the publication. We have moved the lines 331-340 and Figure 9 to the results section. Additionally, we extended this information with the recommendations of the first reviewer.
Round 2
Reviewer 1 Report
Thank you for responding to my comments and revising your manuscript. I believe the additional information and clarifications in your manuscript help improve it, but I still have additional concerns and questions about your methods and the results being presented. Please see my comments below.
Major comments Response:
1. Thank you for adding more information on the experimental design. It is unclear from your description exactly how measurements were extracted for comparison. It seems you recorded 30 seconds of data and then compared the measurement in each second between the vital sign monitor and imaging PPG method. Is that understanding correct? This would seem inappropriate for respiratory rate data which would not be expected to change every second. This may add significant correlation between repeated measurements from the same subject. Please clarify the approach for comparing measurements and provide a rationale for why it is appropriate considering the time-scale that measurements are expected to change over.
2. Thank you for adding the scatter plots. This information helps to show the range of measurements covered by your analysis. I am concerned that these measurements show data below your cut-off frequencies for both heart rate and breathing rate. As suggested in Major Comment 3, the filter cut-off frequencies you selected do not seem broad enough to cover the range of expected physiologic variability. For example, on heart rate you show many data points below 110 BPM while your low cut-off is stated to be 110 BPM. On respiratory rate, many data points are shown below 30 breaths per minute, while your cut-off rate is stated to be 30 breaths per minute. My understanding is that you are attempting to measure frequency content in a range that you are filtering out. Please consider re-performing the analysis with appropriate cut-off frequencies, or explain how your method is accounting for this.
You should include Y-labels on all 4 plots in Figure 6.
Thank you for adding the statistical analysis table, but there is no description of how these values were computed. Specifically you should address who the confidence intervals consider repeated measurements between subjects. I recommend you add a statistical analysis section to your methods to describe how this table and all other analyses reported in the manuscript were determined.
3. See comment above. The use of cut-off frequencies with in the range of the physiologic measurements appears to be a methodological flaw that needs to be corrected. If it is not, then you should provide a better description of your method or analyses that makes it clear why these cut-off frequencies are appropriate for the range of measurements you are using.
4. It is still unclear how your performed the computational time comparison. You refer to CPU usage, but there is not information on the CPU or software methods used to compute this. I recommend considering methods for evaluating the computational costs that do not rely on the CPU used.
You should include Y-labels on Figure 6.
5. Thank you for reorganizing the manuscript.
Minor Comments Response:
Table 2 – Please add the sample size to this table. It is unclear what upper and lower limit are referring to. You should add the units of measurement to HR and BR.
Table 3 – Please clarify how the sample sizes were determined for heart rate and breathing rate and why they are different. You should include units of measurement where appropriate.
Figure 10 – You seem to be indicating that Figure 10 was captured with your method, but the significance of this is still unclear to me. For example, was this same event also captured on the vital signs monitor? That information would help to confirm that it was a real physiological event that your method was able to capture.
Author Response
RESPONSE TO REVIEWER 1
Comments and Suggestions for Authors:
Thank you for responding to my comments and revising your manuscript. I believe the additional information and clarifications in your manuscript help improve it, but I still have additional concerns and questions about your methods and the results being presented. Please see my comments below.
Authors’ Reply: Thanks for your revisions. Few critics do such a professional job. We hope to clarify all your doubts and concerns.
Major Comments:
Major 1 Response
Thank you for adding more information on the experimental design. It is unclear from your description exactly how measurements were extracted for comparison. It seems you recorded 30 seconds of data and then compared the measurement in each second between the vital sign monitor and imaging PPG method. Is that understanding correct? This would seem inappropriate for respiratory rate data which would not be expected to change every second. This may add significant correlation between repeated measurements from the same subject. Please clarify the approach for comparing measurements and provide a rationale for why it is appropriate considering the time-scale that measurements are expected to change over.
Authors’ Reply: Thanks for your comment; we believe that you have a little confusion. We agree that it would be inappropriate to correlate the respiratory rate every second. This would add a greater correlation. You can see table 3, we have detailed a sample size of 360 for the heart rate and 120 for the respiratory rate. This clearly indicates that samples for breathing are taken every 3 seconds. In any case, we have added this information to Table 2 to avoid confusion.
Major 2 Response.
a) Thank you for adding the scatter plots. This information helps to show the range of measurements covered by your analysis. I am concerned that these measurements show data below your cut-off frequencies for both heart rate and breathing rate. As suggested in Major Comment 3, the filter cut-off frequencies you selected do not seem broad enough to cover the range of expected physiologic variability. For example, on heart rate you show many data points below 110 BPM while your low cut-off is stated to be 110 BPM. On respiratory rate, many data points are shown below 30 breaths per minute, while your cut-off rate is stated to be 30 breaths per minute. My understanding is that you are attempting to measure frequency content in a range that you are filtering out. Please consider re-performing the analysis with appropriate cut-off frequencies, or explain how your method is accounting for this.
Authors’ Reply: Once again, we appreciate your insightful comments. Actually, we agree that relaxing the limits would be convenient. This is already detailed in the response to your major comment 3. Obviously, there will be outside measurements to the established limits. We must not forget that the filters are not rigid. As we all know, an ideal filter does not exist. Only, an ideal filter will cancel unwanted frequencies. Real filters have a margin of attenuation. For this reason, we have some measurements outside the limits.
b) You should include Y-labels on all 4 plots in Figure 6.
Very good observation, the Y-labels are not in the PDF file. This problem occurs when generates the PDF. The DOCX file has all the tags. We will be more careful when generating the PDF.
c) Thank you for adding the statistical analysis table, but there is no description of how these values were computed. Specifically you should address who the confidence intervals consider repeated measurements between subjects. I recommend you add a statistical analysis section to your methods to describe how this table and all other analyses reported in the manuscript were determined.
Fortunately, all these estimates can be arrived at by means of existing statistical package such as MedCalc® that directly incorporate a procedure for comparing variables. In this calculation software were implemented the correlation coefficients, comparison, and graphical methods, without the algebraic manipulation of the variables being necessary.
In any case, your recommendation is important. We have added the following sentences:
“In addition, the estimated and reference data were correlated. Respectively, 360 and 120 pairs of measurements from 9 videos for HR and BR are plotted in Figure 6 (c) and (d).”
“All the statistical analysis, tables and graphical methods have performed with the MedCalc® statistical package.”
Finally, we respect and appreciate your comment. We consider important his recommendation that we have to correlate the measurements. However, we also think that the correlation coefficient measures the relationship between two variables (how one varies when the other varies), but not the degree of agreement. The latter, if done through Bland-Altman. In any case, we consider that the dispersion diagram provides important information about the experiment.
Major 3 Response.
See comment above. The use of cut-off frequencies with in the range of the physiologic measurements appears to be a methodological flaw that needs to be corrected. If it is not, then you should provide a better description of your method or analyses that makes it clear why these cut-off frequencies are appropriate for the range of measurements you are using.
Authors’ Reply: Thank you for your comment, but this comment would be answered with the main answer 2. We continue to believe that the limits should relax. However, as we already explained, the real filters are attenuated in the limits. These limits were established by the doctor and we respect his medical judgment. Finally, the measurements are showing that there is a correlation.
Major 4 Response.
a) It is still unclear how your performed the computational time comparison. You refer to CPU usage, but there is not information on the CPU or software methods used to compute this. I recommend considering methods for evaluating the computational costs that do not rely on the CPU used.
Authors’ Reply: As always, your observation is very successful. We have clarified the paragraph; we hope to clarify your doubts.
“CPU usage was recorded when both scripts worked at the same time. Scripts analyzed and processed the same information. In order to show the lightweight of our algorithm, we have registered the CPU usage history through data comparison graph. The computer was an Intel(R) Core(TM) i7-4500U CPU (1.8 GHz, 4 MB cache)”.
b) You should include Y-labels on Figure 6.
Authors’ Reply: It was already answered, thanks.
Major 5 Response.
Thank you for reorganizing the manuscript.
Authors’ Reply: Your feedback was very valuable. Our paper improved significantly.
Minor comments 1
Table 2 – Please add the sample size to this table. It is unclear what upper and lower limit are referring to. You should add the units of measurement to HR and BR.
Authors’ Reply: Ok, we have added the sample number. The upper and lower limits are the concordance limits values.
Minor comments 2
Table 3 – Please clarify how the sample sizes were determined for heart rate and breathing rate and why they are different. You should include units of measurement where appropriate.
Authors’ Reply: We have added the following paragraph:
“In the HR case, the measurements are recorded every second. In contrast, the BR is recorded every three seconds, thus an over correlation is avoided by repetition of values.”
Minor comments 3
Figure 10 – You seem to be indicating that Figure 10 was captured with your method, but the significance of this is still unclear to me. For example, was this same event also captured on the vital signs monitor? That information would help to confirm that it was a real physiological event that your method was able to capture.
Authors’ Reply: Very good appreciation, this is something that was discussed with a pediatrician. We have plotted NICU monitor values. We did not want to plot the monitor values because the figure would not be our development. However, we respect your criteria; our role has significantly improved with your contributions.
Figure 10. Signal Monitor development in Matlab vs NICU monitor.
Reviewer 4 Report
Fig. 6, there is no description of the y-axis.
The language needs to be further polished.
Author Response
RESPONSE TO REVIEWER 4
Comments and Suggestions for Authors:
Comment 1
Fig. 6, there is no description of the y-axis.
Authors’ Reply: We do agree with your appreciation of this point, but we have followed the recommendation of another reviewer and we have completed the figure. This was a problem with PDF generator.
Comment 2:
The language needs to be further polished.
Authors’ Reply:
We do agree that sometimes the language barriers prevent from expressing correctly the ideas and making them easily understandable, and that is why we contracted a professional proofreader to help us improve the language of this contribution. We are struck by your comment.